# Species richness estimation of the Afrotropical Darwin wasps (Hymenoptera, Ichneumonidae)

Noah Meier [1]*, Mikhaila Gordon [2], Simon van Noort [2,3], Terry Reynolds [2,4], Michal Rindos [5,6], Filippo Di Giovanni [7], Gavin R. Broad [8], Tamara Spasojevic [1,9,10], Andrew Bennett [11], Davide Dal Pos [12], Seraina Klopfstein [1,9]

1 Naturhistorisches Museum Basel, Basel, Switzerland, 2 Research and Exhibitions Department, South African Museum, Iziko Museums of South Africa, Cape Town, South Africa, 3 Department of Biological Sciences, University of Cape Town, Cape Town, South Africa, 4 Agricultural Research Council–Plant Health and Protection, Biosystematics, Pretoria, South Africa, 5 Biology Centre of the Czech Academy of Sciences, Institute of Entomology Ceske Budejovice, Czech Republic, 6 Faculty of Science, University of South Bohemia, Ceske Budejovice, Czech Republic, 7 Department of Life Sciences, University of Siena, Siena, Italy, 8 The Natural History Museum, Cromwell Road, London, United Kingdom, 9 University of Bern, Institute of Ecology and Evolution, Bern, Switzerland, 10 Naturhistorisches Museum Wien, Wien, Austria, 11 Canadian National Collection of Insects, Arachnids and Nematodes, Agriculture and Agri-Food Canada, Ottawa, Ontario, Canada, 12 Department of Biology, University of Central Florida, Orlando, Florida, United States of America

* noah.meier@bs.ch

**Data Availability Statement:** All relevant data are within the manuscript and its Supporting Information files.

## Abstract

Species richness is one of the fundamental metrics of biodiversity. Estimating species richness helps spotlight taxonomic groups that are particularly under-studied, such as the highly diverse Darwin wasps. The only available estimate of the number of Darwin wasps in the Afrotropics proposed almost 11,000 species, compared to the 2,322 recorded species. However, it relied exclusively on the ratio of morphospecies to described species in Henry Townes' personal collection. We provide an updated estimate of the Afrotropical Darwin wasp species, using empirical data from multiple sources, including the increase in species numbers following generic revisions, morphospecies sorting in natural history collections, and diversity patterns of better-studied insects (butterflies) for extrapolation. Our analyses suggest that our knowledge of Darwin wasps is highly incomplete, with only 13–22% of species known in the five most extensively studied countries in the Afrotropics. We estimate 9,206–15,577 species of Darwin wasps within the entire Afrotropics, with the highest concentration expected in the Equatorial Afrotropics and Madagascar. Due to data constraints, our approach tends to underestimate diversity at each step, rendering the upper estimate (15,577 species) more realistic. We highlight reasons contributing to the gap between recorded and estimated species richness, including logistical and financial factors, as well as post-colonial influences.

**Funding:** SK was supported by grant 310030_192544 of the Swiss National Science Foundation. The funder played no role in study design, data collection and analysis, decision to publish, or preparation of the manuscript.

**Competing interests:** The authors have declared that no competing interests exist.

## Introduction

### Species richness—A crucial metric for biodiversity management

Biodiversity, including species richness, genetic diversity and the diversity of ecosystem functions, is of fundamental value for nature itself but also for human well-being [1]. The study of taxonomy and species boundaries serves as a crucial tool in comprehending the network of ecosystem services. Among the metrics used, species richness stands out as the most commonly employed indicator of biodiversity. However, except for the best-studied groups such as vertebrates [2] and flowering plants [3], we have still only described a small proportion of the species, and are thus relying on estimates to predict their diversity [4, 5].

Approaches to estimating species richness usually combine empirical data with logical assumptions or simplistic models, such as the known number of species relative to rates of description [6]; the relation between higher-rank and lower-rank taxa [7]; ecological relationships [8, 9]; or taxonomic effort [10]. Although the resulting estimates of global species richness vary widely [11], they all agree on the fact that we have only described a fraction of the true diversity [7–10, 12–16]. Together with nematodes and mites, insects appear to have the largest ratio of undescribed to described species within Metazoa [7].

While Erwin [8] estimated 30 million species of tropical arthropods by extrapolating the number of host-specific beetle species found on the tree *Luehea seemannii* Triana & Planch to all tropical tree species, more recent estimates usually range around 3–7 million species [5, 6, 11, 12, 17, 18]. This is several times more than the known 1.1 million insect species [19]. Within insects, the gap is thought to be particularly large in Diptera and parasitoid Hymenoptera [20–23].

### Parasitoid wasps—Mind the gap

The true diversity of parasitoid wasps is arguably one of the big unknowns in current biodiversity research. Estimates based on host-parasitoid-ratios suggest that there could be as many as 0.8–1.1 million species [9]. Undoubtedly, the number of undescribed parasitoid wasps ranges in the hundreds of thousands [12, 24]. The majority of these are likely to be found in the megadiverse superfamilies Chalcidoidea and Ichneumonoidea [12, 24]. Studying the systematics and ecology of parasitoids is not only a valuable contribution to the scientific community, but also economically of great importance, due to their lifestyle and thus potential as biocontrol agents for pest insects. The fact that they are drastically under-studied complicates the strategic application of parasitoids in organic agriculture. Furthermore, species richness estimations that intend to quantify and highlight the magnitude of undescribed species within these groups are often based on extrapolations from very limited data, or they entirely reflect expert opinions [25].

In braconid wasps (Braconidae), one of the two extant families included in Ichneumonoidea, there are currently 22,800 described species [19, 26], and their global species richness has been estimated using several different approaches [6, 25, 27]. Despite varying methodological approaches, the results are rather consistent. Dolphin & Quicke [6] estimated 30,000–50,000 braconid species. The lower number was reached by fitting, for each subfamily, a logarithmic function to the description rate versus the currently described species. This function was extrapolated to establish an estimate of the total number of species. The higher number was obtained by, again for each subfamily, calculating the ratio of braconid to butterfly or to mammal species in the Western Palaearctic and extrapolating these to obtain the global species richness. Depending on which surrogate group was used, the results varied by about a factor two [6]. Jones et al. [27] estimated 42,600 braconid species by compiling the ratios of the number

of species before and after taxonomic revisions. These ratios were then applied to all the unrevised genera [27]. Since this approach tends to underestimate species richness, as generic revisions are hardly ever complete or final, Jones et al. [27] used a non-parametric species richness estimator (Chao-I) to account for rare species that had not yet been collected.

In Darwin wasps (Ichneumonoidea, Ichneumonidae), there are currently 26,200 species described [19, 26]. Henry Townes estimated the global species richness of Darwin wasps by extrapolating the ratio of described to undescribed morphospecies in his collection to the known species richness for each geographic realm. His analysis resulted in a rough estimate of 60,000 species worldwide [28]. This number has been deemed too low by experts in subsequent publications, but apart from some rough guesses [29, 30], no attempt has been undertaken to update that number in more recent times. In the Western Palaearctic, the number of described species since Townes made his estimate already exceeds his prediction for the region by more than 1,400 species or roughly 25% [26, 28], although several very large genera still await taxonomic revision. In contrast, in realms where less taxonomic effort has been made [6], the number of described species still lags far behind the regional estimates of Townes. This is especially the case in the Afrotropical region, which records 2,070 species compared to 10,787 estimated species [26, 28]. It remains to be determined how much of the species lag in the Afrotropical region is attributable to study biases, such as a limited number of taxonomists working in this area, and challenges with respect to accessibility of localities, or whether Townes [28] merely overestimated the diversity of Darwin wasps in this realm.

## Latitudinal distribution anomaly

For most groups of organisms, the recorded (or estimated) species richness in tropical regions exceeds that of temperate regions [31], a pattern that seemed not to apply to Darwin wasps, at least at first. Owen & Owen [32] proposed that they instead showed an anomalous species distribution with a higher diversity in the temperate regions, based on the catches of a small number of Malaise traps put up in the United Kingdom and in Sierra Leone. This hypothesis led to numerous subsequent studies and theories aiming to explain the observed phenomenon, such as the species fragmentation theory, which argues that the numerous tropical insect species are each too rare to sustain a host-specific parasitoid wasp species [33, 34]. However, these studies built their hypotheses on geographically limited sampling approaches, with a particular deficiency in sampling effort in equatorial diversity hotspots [35]. More recent extensive sampling, especially in the Neotropical region, has called the latitudinal gradient anomaly of ichneumonids into question [36–39], but our still poor knowledge of the diversity of the family at low latitudes makes a definite conclusion on this matter difficult [25]. In the Afrotropical region, extensive sampling efforts are still rare, and the true diversity of Darwin wasps thus remains unclear [40].

## Estimating Afrotropical Darwin wasp diversity

In this paper, we provide a new estimate for the species richness of Afrotropical Darwin wasps, and at the same time identify the largest gaps in our knowledge. In an attempt to account for data deficiency, we combine several previously utilised approaches in a stepwise estimation procedure. In contrast to the Townes estimate [28], which was derived from a single collection, we integrate empirical data from multiple sources, including generic revisions, morphospecies of selected genera in two natural history collections, and distribution patterns of a more intensively studied insect group, the butterflies. Furthermore, we try to identify biases due to uneven taxonomic effort, uneven coverage of geographical areas, and poor data availability. Finally, we identify the areas with the most urgent need of taxonomic effort and argue that the gaps in our

knowledge can only be filled through intensified collaboration, as reference collections are often split between museums of former colonial powers and the countries of origin.

## Materials and methods

### General approach

We subdivided the Afrotropical realm into five ecogeographical subrealms, each represented by one focal country. Then, we used a stepwise procedure to estimate the number of Darwin wasp species in the Afrotropical region (Fig 1). We here give an overview of these steps, with details following further below. **Step 1**: We used generic revisions and sorted morphospecies in collections to extract factors that describe the increase in the number of species in the treated genus. We further accounted for rare species that have not yet been collected by incorporating a Chao-I estimator that is based on the number of species represented by only one or two specimens into these factors. This resulted in mean multipliers obtained for each subrealm. **Step 2**: These multipliers were then used to estimate the species richness in the five focal countries by extrapolating the number of species in the genera not yet revised in each

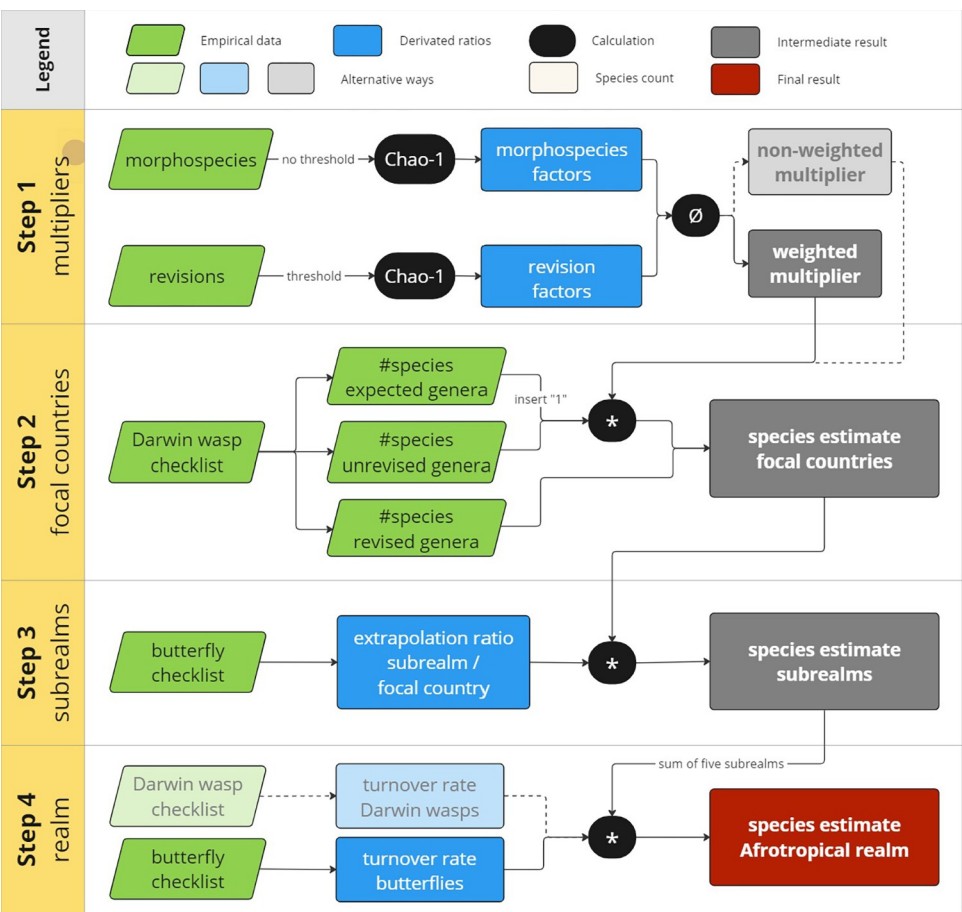

**Fig 1. Workflow illustration.** Steps 1–3 are done separately for each subrealms. Green fields represent empirical data. White field represent species counts and blue fields represent derived ratios. Black fields are calculations. Grey fields are subresults and the red field represents the final result; # stands for "number of", * stands for multiplication, Ø stands for a mean calculation. "insert 1" indicates that we used "1 species" as minimal representative for expected genera in the focal countries. Light colored fields and dotted arrows represent alternative ways for calculation.

subrealm. **Step 3**: Since very little is known about Darwin wasps beyond these focal countries, we used the species richness of butterflies in each country as a proxy for the diversity of its ecological niches. Thus, we extrapolated the species richness of Darwin wasps from the focal countries to the subrealms. **Step 4**: To account for overlap between the species present in the five subrealms, we multiplied the cumulative number of Darwin wasp species with the turnover rate calculated for butterflies between the five subrealms. The complete methodological pipeline is available as an R-script in S1 File.

## Ecogeographical subdivision

Due to extremely limited data on the diversity of Darwin wasps in most Afrotropical countries, we had to limit the number of geographical units for species richness estimation. Particularly in Darwin wasps, diversity assessments make much more sense based on larger geographical units, since many species range across whole or even multiple realms [26, 41]. We thus treated the Afrotropical realm as subdivided into five subrealms following the Bioregions 2020 proposal by One Earth [42]. These bioregions are based on plant and zoogeographic distribution patterns and incorporate multiple factors, such as human land use and canopy cover [42–44]. For each of the subrealms (A–E), we selected a single focal country that had at least 200 described species of Darwin wasps, to be considered relatively better-known in terms of Darwin wasp faunistics (Table 1, Fig 2). Note that even though these countries have the highest recorded number of species, they are still severely understudied. For the Equatorial Afrotropics, there are two countries qualifying as focal countries: Uganda (N = 294) and the Democratic Republic of Congo (N = 340). We used the latter as a focal country in this study due to the higher number of recorded species.

## Step 1 –Revision factors, morphospecies factors and multipliers

**Revision factors.** Similar to the approach in Jones *et al.* [27], we calculated the ratio between the number of species within a genus after and before a taxonomic revision. We included revisions since the year 1967, since this marks the more recent history of Darwin wasp research, initiated by the extensive revisions of Heinrich [45–48] and the Townes volumes [28, 49–51] establishing a generic classification for most Ichneumonidae. We only considered articles here that studied at least as many specimens as there were species described for the genus in the Afrotropical region. If there were several revisions for the same genus

**Table 1. Assignment of Afrotropical countries to subrealms A–E.**

| Subrealm | Name | Focal country | Included countries (ISO 3166 3-letter) |
|---|---|---|---|
| A | Sub-Saharan Afrotropics | Kenya | BEN, BFA, TCD, ERI, ETH, GAB, KEN, MLI, MUS, NGA, RWA, SEN, SOM, SDN, TGO, YEM, Soc, SAU, (CAF, GMB, GHA, GIN, CIV, MOZ, NER, TZA, UGA) |
| B | Equatorial Afrotropics | Democratic Republic of Congo | CAF, GMB, GHA, GIN, CIV, NGA, UGA, BDI, CMR, COG, COD, GNQ, GNB, LBR, SLE, Asc, StH, STP (MOZ, TZA, BEN, KEN, SEN, TGO, AGO, MWI, ZMB) |
| C | Sub-Equatorial Afrotropics | Tanzania | MOZ, TZA, AGO, MWI, ZMB, ZWE (KEN, COD, BWA, NAM) |
| D | Southern Afrotropics | Republic of South Africa | BWA, NAM, ZAF, LSO, SWZ (AGO) |
| E | Madagascar & East African Coast | Madagascar | MDG, COM, MUS, MYT, REU, SYC (SWZ, MOZ, TZA, SOM) |

Countries that cover several subrealms are assigned to the one covering the larger proportion of the countries' area and the others are mentioned in brackets. Focal countries have at least 200 species of Darwin wasps reported. Ascension Island (Asc), Saint Helena (StH) and Socotra (Soc) have either no or no separate ISO3166 3-letter codes, thus the abbreviations in brackets were used.

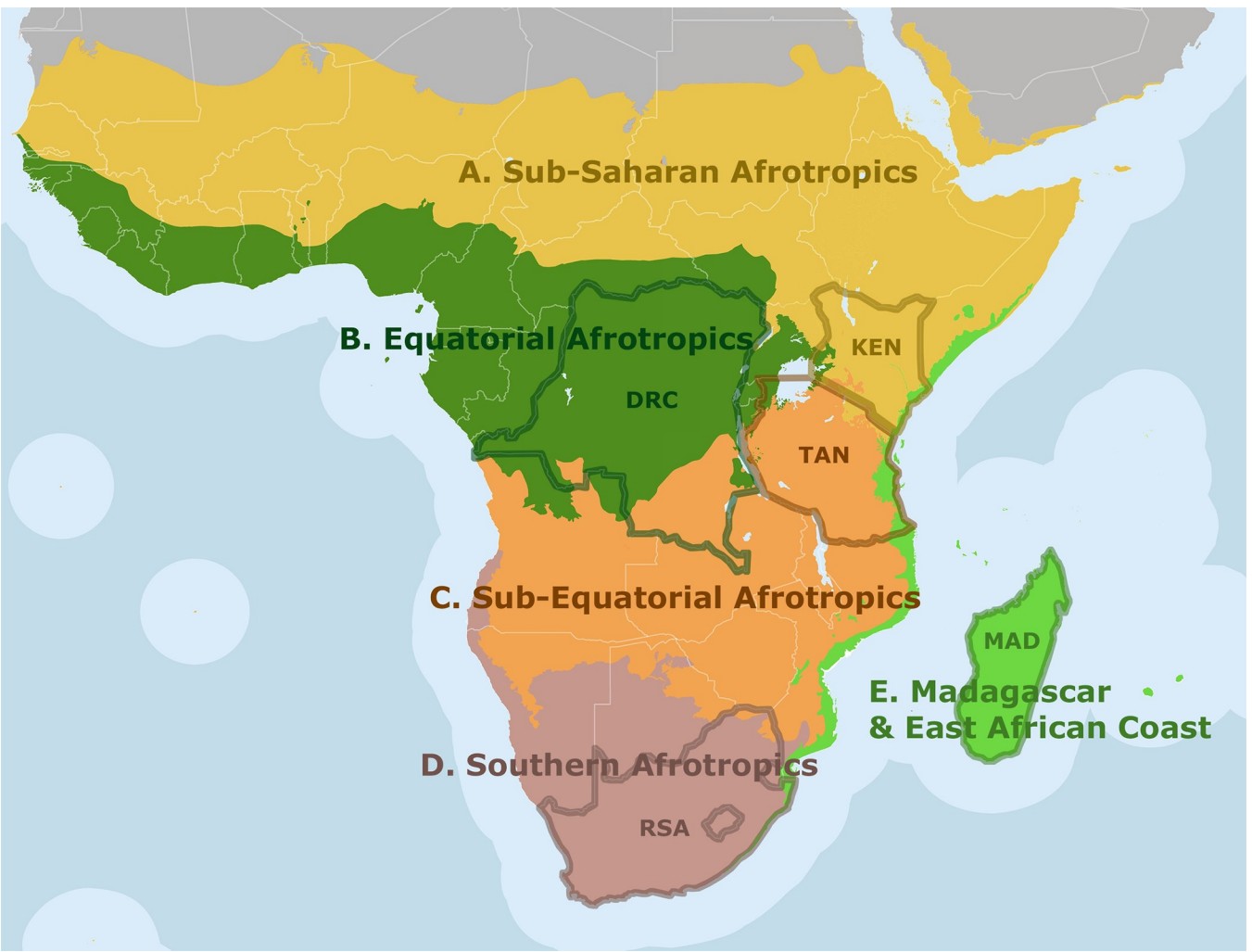

**Fig 2. Division of the Afrotropical realm into five subrealms.** A, Sub-Saharan Afrotropics; B, Equatorial Afrotropics; C, Sub-Equatorial Afrotropics; D, Southern Afrotropics; E, Madagascar & East African Coast. Focal countries are framed in red: KEN, Kenya; DRC, Democratic Republic of Congo; TAN, Tanzania; RSA, Republic of South Africa; MAD, Madagascar. Reprinted from [42] under a CC BY license and adapted with permission from One Earth, ©One Earth 2020.

matching the requirements, we extracted the data for all the revisions cumulatively. We treated revisionary taxonomic data from 49 articles, covering 105 of the 369 genera of Afrotropical Darwin wasps.

We detected large heterogeneity in the scope of the studied revisions, ranging from single species descriptions based on a handful of specimens, to very comprehensive revisions including hundreds of specimens. Only revisions that approach the actual diversity of a genus, enable extrapolations to unrevised genera. Therefore, we conservatively excluded the 50% of the studies that had the lowest specimen / species ratio. A list of the revisions and the extracted data used for this approach is provided in S2 File.

**Morphospecies factors.** As the taxonomic coverage of the treated revisions was very uneven, we complemented this data by following the Townes [28] approach for estimating species richness in Darwin wasps, by including two collections with morphospecies sorted by H. Townes (Iziko South African Museum, Cape Town) and G. Broad (the Natural History Museum, London) and checking these against described species. The ratio of the number of

undescribed morphospecies and described species per genus in the collection was used to calculate morphospecies factors. For series of morphospecies of known genera consisting only of undescribed species, we inserted "1" as the number of described species to calculate a ratio. Townes' sorting covered 72 genera from 15 subfamilies only from South Africa of which Cryptinae, Campopleginae and Phygadeuontinae were the most abundant. This was complemented by Broad's sorting of 8 genera of Orthocentrinae from various locations. A list with all morphospecies data is provided in S2 File.

**Chao-I.** Generic revisions and morphospecies sorting can obviously only use species that have already been collected. We used a non-parametric species richness estimator (Chao-I) to account for rare species that have not yet been collected [52]. The Chao-I estimator appraises the total number of species, including the undetected ones, based on the number of species represented by only one (singleton) or two (doubleton) specimens [52]. Chao-I was calculated in a bias-corrected form that allows the number of doubletons to be equal to zero [53]:

$$S_{est} = S_{obs} + \frac{singletons \times (singletons - 1)}{2 \times (doubletons + 1)} \tag{1}$$

Where $S_{est}$ is the estimated number of species and $S_{obs}$ is the observed number of species, which is equivalent to the sum of described and undescribed species in the collection (morphospecies factors) or species after a revision (revision factors).

**Multipliers.** All revision and morphospecies factors were then assigned to one or several subrealms reflecting the origin of the material studied. The effects of revision and morphospecies factors were tested separately. However, since these factors represent complementary data, we decided to use the mean across both factors as multipliers for this analysis.

From museum collections, it is apparent that species-rich genera are for the most part unrevised (with some exceptions, e.g., *Enicospilus* Stephens [54]; G. Broad pers. obs.), which likely reflects the preference of taxonomists for first treating groups of a size that is feasible to handle in a reasonable amount of time. However, the species-poor genera not only likely show more variance due to small size, but also likely show a smaller increase in species numbers after revisions. To prevent overweighting genera with few species, we also calculated weighted multipliers as weighted means, with the weights reflecting the number of species in a particular genus. We use the span between the calculations based on non-weighted and weighted multipliers to suggest a range in our estimates.

## Step 2 –Estimating the species richness in the focal countries

For each subrealm, the individually calculated non-weighted and weighted multipliers were used to estimate the number of species in genera that were not yet revised. We used a revised checklist of Afrotropical Darwin wasps as reference for analysis (S3 File). We found that most Afrotropical Darwin wasp genera without revision were not even recorded yet in the focal countries, despite most of them showing very wide distributions in the rest of the world. Therefore, if a genus without records in the focal country was recorded in any of the other countries of the subrealm, we also expected it to be present in the focal country. We accounted for these expected genera very conservatively by adding 1 * multiplier to the species richness estimate. For revised genera, we took the non-extrapolated number of recorded species to estimate the species richness in the focal countries.

## Step 3 –Faunistic surrogates

Using better investigated taxonomic groups, such as butterflies, as surrogates to estimate the diversity of less studied taxa, such as parasitoid wasps or flies, has proven to be very useful [6].

With the application of butterflies as surrogates, we assumed that species richness patterns are similar among taxonomic groups, as they all depend primarily on the diversity of ecological niches. In addition to reflecting the richness of ecological niches in an area for flying insects in general, butterflies are frequent hosts for this family of parasitoids [55] and thus represent different ecological niches themselves. However, as butterflies with their herbivorous larvae adopt a life strategy that is very different from parasitoid wasps, it is likely that the relation between the number of Darwin wasps and butterflies varies between different ecoregions. Therefore, we estimated the relationship between species richness of Darwin wasps and butterflies for each subrealm independently. The distribution of butterflies in the Afrotropical realm was extracted and updated from a public database [56] (S4 File), and we assumed that the ratio between species richness in the subrealm and the corresponding focal country is the same for butterflies and Darwin wasps.

## Step 4 –Turnover between subrealms

Like butterflies, most Darwin wasps are winged insects and considered highly mobile [41]. Therefore, we assumed that both groups show a similar turnover rate across the five subrealms. We calculated the turnover rate as the ratio between the distinct number of described species across all subrealms and the cumulative number of species in the entire Afrotropical realm. A high turnover rate indicates littleoverlap in the species assemblage across the subrealms and a low turnover rate indicates extensive overlap. We calculated the turnover rate for Darwin wasps and butterflies, independently. However, we decided to use the turnover rate calculated based on the distribution of butterflies, since the turnover rate of Darwin wasps is likely to be too large due to data deficiency in the study of distribution ranges. Hence, the total number of Afrotropical Darwin wasp species was estimated by multiplying the cumulative number of Darwin wasp species across the five subrealms with the turnover rate.

## Results

### Revision and morphospecies factors & multipliers

Among the 105 genera revised in the literature, we found a median ratio of specimens studied per species in a genus of 3.65 (Fig 3). We used this threshold to conservatively exclude half of the studies as being based on an insufficient number of species to even approach their true diversity. After this filter, there were still between 25–31 generic revisions available for calculating revision factors for each of the five subrealms. In contrast, morphospecies data was obtained only for 2–7 genera (of Orthocentrinae) per subrealm, except for the Southern Afrotropics, where additional material from Townes' sorting at the Iziko South African Museum was available for 72 genera.

Revision factors (average multiplier 4.64; average weighted multiplier 6.47) were distinctly lower than morphospecies factors (average multiplier 20.21; average weighted multiplier 42.09) across the subrealms (Fig 4). The multipliers used in the downstream analysis were based on both revision factors and morphospecies factors and were intermediate compared to the multipliers based on a single type of factors (average multiplier 6.32; average weighted multiplier 11.32). Furthermore, weighted multipliers were about 1.5–2 times higher than (non-weighted) multipliers. The complete set multipliers for each subrealm is given in S5 File.

### Focal countries

We found in all focal countries 67–157 (median = 100) recorded genera without taxonomic revisions and 4–101 (median = 22) genera that were expected to occur in the focal country due

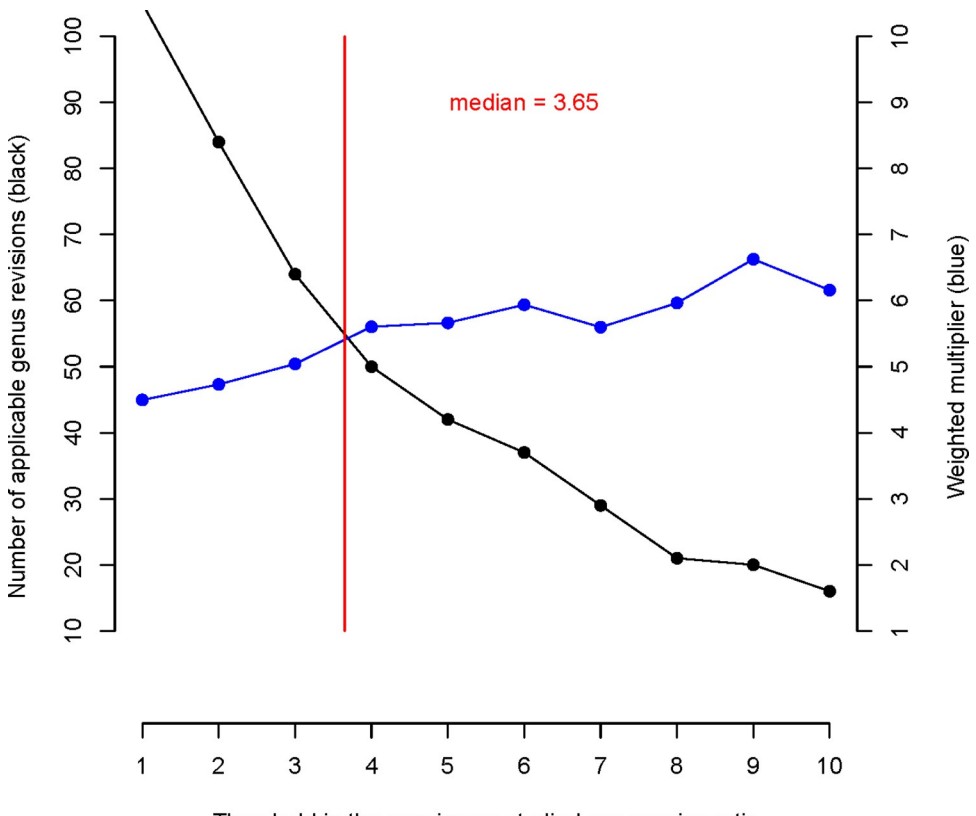

**Fig 3. Relation between the weighted multiplier and the threshold in the number of specimens studied per species ratio for the entire Afrotropics.** The weighted multipliers (blue dotted line, right Y-axis) are only based on revisions. The number of applicable genera (black dotted line) corresponds to the left Y-axis. The median threshold (3.65 specimens per species) is indicated by the red vertical line.

to records in other countries of the subrealm (Table 2). The Democratic Republic of Congo has both the lowest number of genera without taxonomic revision and the highest number of expected genera (Table 2).

The estimated species diversity for the five focal countries ranged from 1,498 (Kenya) to 3,082 (Madagascar) using non-weighted multipliers, and from 2,543 (Kenya) to 5,407 (Madagascar) using weighted multipliers (Table 2). This suggests that on average, only between 13–22% (weighted–non-weighted multiplier) of the species richness of Darwin wasps is currently known in the focal countries, and it is highly likely that the proportion is much lower in all the other countries.

The estimated diversity per subfamily varies substantially between focal countries (Fig 5). Regarding current species records, Ichneumoninae and Ophioninae are the most diverse in all focal countries. However, our analysis suggests Ichneumoninae, Cryptinae and Pimplinae as the three most species-rich subfamilies, of which Ichneumoninae are expected to be the most species-rich in all focal countries except for Madagascar, where Phygadeuontinae reach even higher estimates. Ophioninae gain only few additional species in our estimation since most genera were already considered taxonomically revised [54].

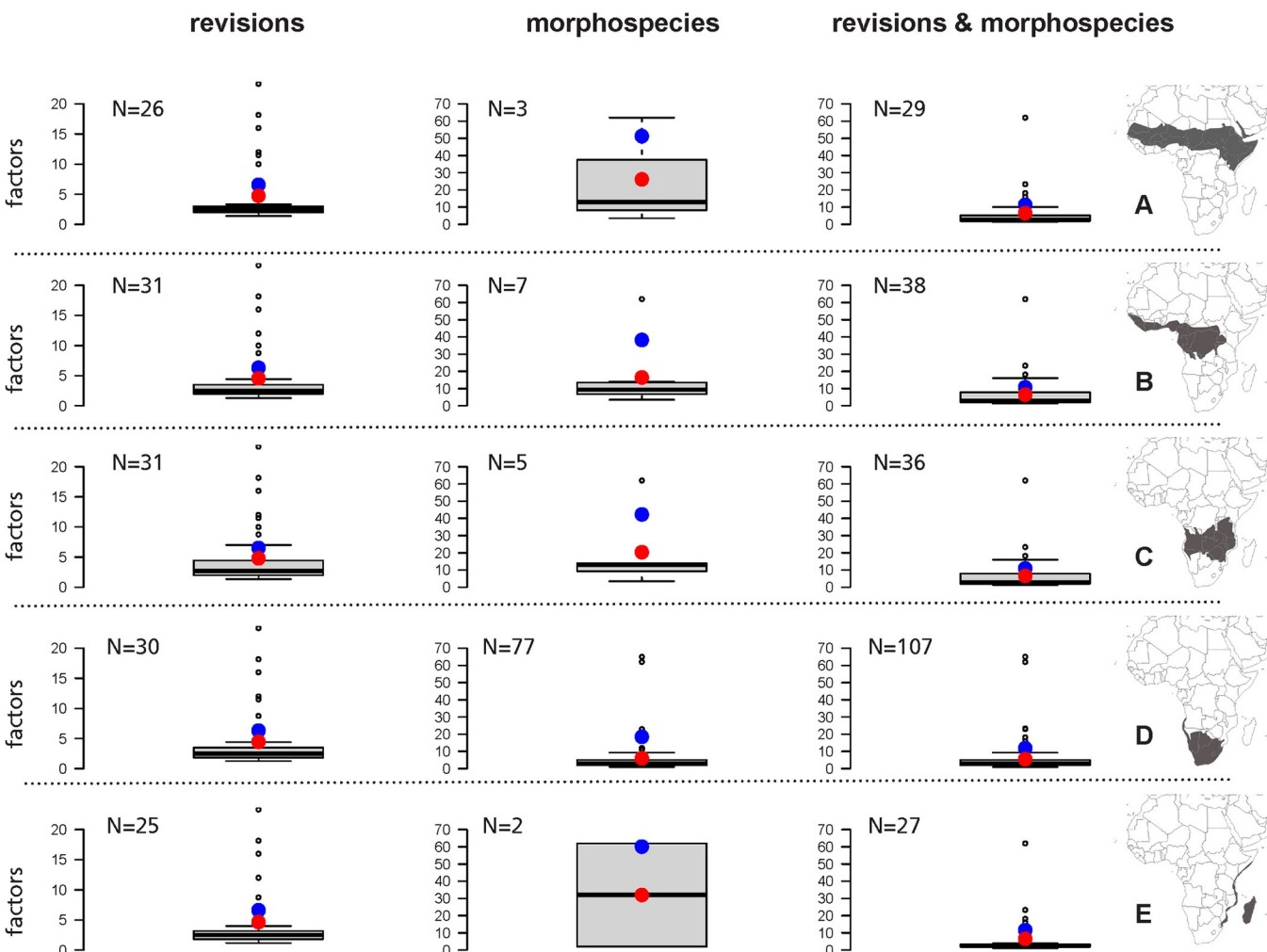

**Fig 4. Distribution of revision and morphospecies factors and the combined data for each subrealm.** A, Sub-Saharan Afrotropics; B, Equatorial Afrotropics; C, Sub-Equatorial Afrotropics; D, Southern Afrotropics; E, Madagascar & East African Coast. The blue dots indicate the weighted multipliers (weight = number of species in a genus) and the red dots indicate the non-weighted multipliers. N indicates the number of genera in each chart. All the calculated multipliers are given in S5 File.

**Table 2. Species richness estimation of Darwin wasps in the focal countries of the five Afrotropical subrealms using multipliers derived from revisions and morphospecies.**

| Focal country | KEN | DRC | TAN | RSA | MAD |
|---|---|---|---|---|---|
| Current species richness | 259 | 340 | 482 | 653 | 601 |
| Non-weighted multiplier | 6.53 | 6.45 | 6.60 | 5.52 | 6.47 |
| Weighted multiplier | 11.39 | 10.77 | 10.93 | 11.84 | 11.64 |
| Recorded genera without revisions or morphospecies | 78 | 67 | 100 | 105 | 157 |
| Additional expected genera | 50 | 101 | 22 | 4 | 17 |
| Total estimated Darwin wasp species with non-weighted multiplier | 1,498 | 2,309 | 2,191 | 1,809 | 3,082 |
| Total estimated Darwin wasp species with weighted multiplier | 2,543 | 3,795 | 3,492 | 3,276 | 5,407 |

KEN, Kenya; DRC, Democratic Republic of the Congo; TAN, Tanzania; RSA, Republic of South Africa; MAD, Madagascar.

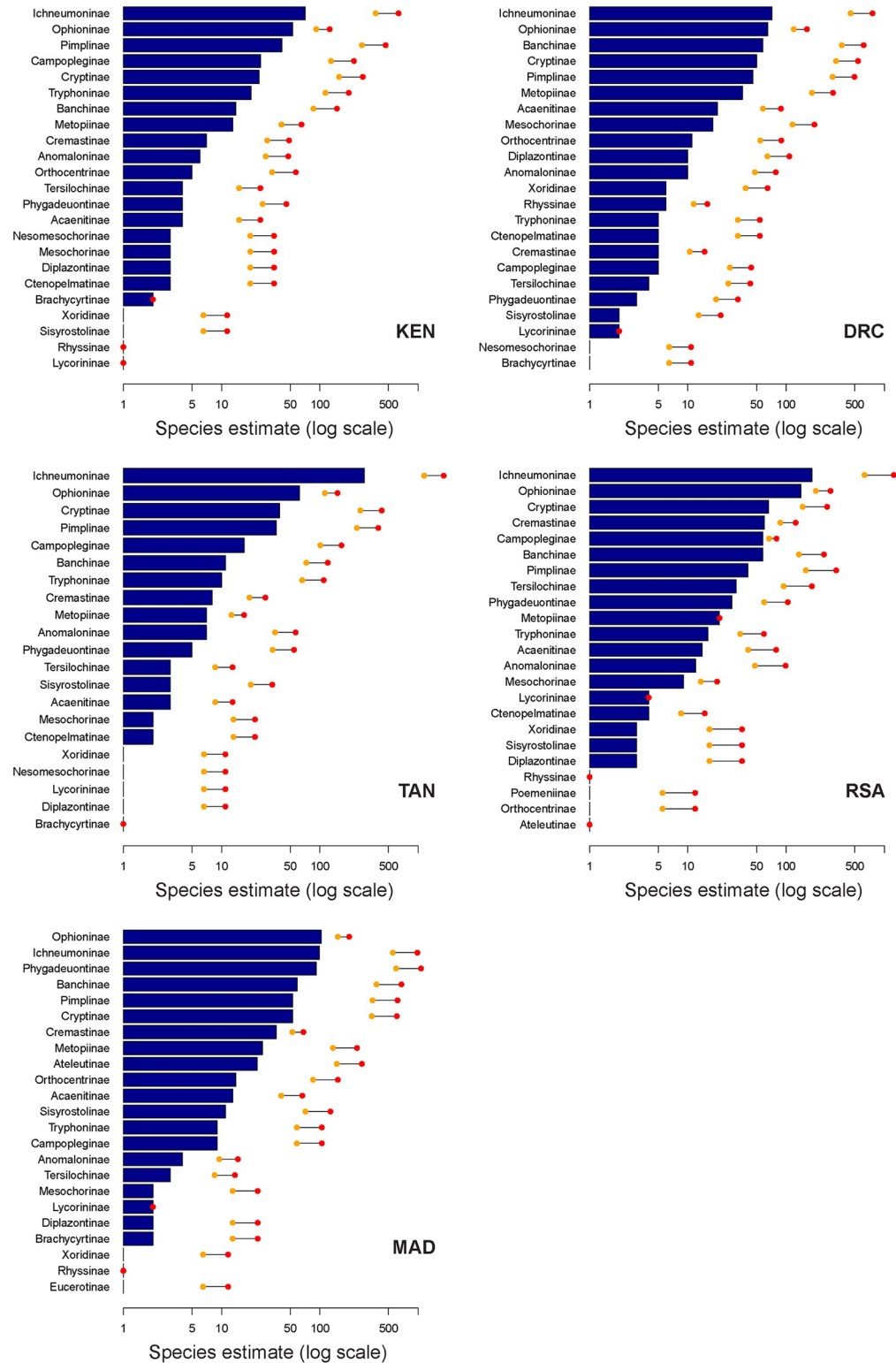

**Fig 5. Estimation of species richness per subfamily of Darwin wasps in the five focal countries.** KEN, Kenya; DRC, Democratic Republic of Congo; TAN, Tanzania; RSA, Republic of South Africa; MAD, Madagascar. Blue bars represent the currently recorded number of species in the country (ordered). The orange and red dots represent the estimated number of species using non-weighted and weighted multipliers, respectively. The X-axis is on a logarithmic scale.

**Table 3. Species richness estimation of Darwin wasps in the five Afrotropical subrealms using butterflies as faunistic surrogates.**

| Subrealm | A | B | C | D | E | Total |
|---|---|---|---|---|---|---|
| Recorded butterfly species in focal country | 912 | 1,813 | 1,329 | 669 | 325 | |
| Recorded butterfly species in subrealm | 1,902 | 3,097 | 1,846 | 748 | 390 | |
| Ratio subrealm / focal country | 2.09 | 1.71 | 1.39 | 1.12 | 1.20 | |
| Total estimated Darwin wasp species with non-weighted multiplier | 3,123 | 3,944 | 3,043 | 2,023 | 3,698 | 15,831 |
| Total estimated Darwin wasp species with weighted multiplier | 5,304 | 6,482 | 4,850 | 3,663 | 6,489 | 26,788 |

Columns represent the five subrealms: A, Sub-Saharan Afrotropics; B, Equatorial Afrotropics; C, Sub-Equatorial Afrotropics; D, Southern Afrotropics; E, Madagascar & East African Coast. The total indicates the cumulative number of species across the five subrealms.

## Butterfly surrogate

We used a public distribution database for the Afrotropical butterflies [56], to extract the number of butterfly species per country and calculate species turnover rates between the focal countries and the corresponding subrealms, and between entire subrealms (Table 3). The ratio of the number of recorded butterfly species in the subrealm and the corresponding focal country ranged between 1.12 (D, Southern Afrotropics) and 2.09 (A, Sub-Saharan Afrotropics; Table 3). By multiplying these ratios with the estimated Darwin wasp diversity of the five focal countries, we calculated the estimated diversity of Darwin wasps for the five subrealms (Table 3): The highest species diversity was obtained for Madagascar and East African Coast (subrealm E: N = 6,489), closely followed by the Equatorial Afrotropics (subrealm B: N = 6,482).

## Species richness of Darwin wasps in the Afrotropical realm

We used the ratio between the distinct number of species of the five subrealms (butterflies: 7,983; Darwin wasps: 3,242) and cumulative number of species in the Afrotropical realm (butterflies: 4,642; Darwin wasps: 2,322) for the currently recorded butterflies and Darwin wasps, respectively, as approximation for the species turnover rates between subrealms. The calculated turnover rate was smaller in butterflies (0.58) than in Darwin wasps (0.71), suggesting a smaller overlap in the species assemblage of Darwin wasps across the subrealms, or more likely an overestimation of the turnover rate due to data deficiency. We thus corrected the cumulative estimated species richness retrieved for the five subrealms with the turnover rates calculated for butterflies, which resulted in an estimate of Darwin wasp species richness in the Afrotropical realm of 9,206–15,577 species.

## Discussion

### Why our estimate is probably too low

Townes [28] estimated a species richness of Darwin wasps in the Afrotropical realm of 10,787 species, which lies slightly above the lower of the estimates obtained in our study, and considerably below the higher one (9,206–15,577 species). In contrast to Townes [28], who based his estimate exclusively on extrapolations from samples in his own collection, we used multiple established methods and empirical data from various sources [6, 27, 28]. Nevertheless, the data gathered to estimate the species richness in the focal countries is based on material in natural history collections that was used for morphospecies sorting or for generic revisions. Besides the obvious limitation that most of the existing specimens have yet to be revised, this material does not represent the diversity of Afrotropical Darwin wasps equally, since many ecotypes and countries have never been effectively sampled, which is a widespread issue in tropical

regions [25, 57]. Besides the five focal countries used in this study, Angola and Uganda are the only other Afrotropical countries which have had any reasonable sampling conducted [26, 58, 59]. Although at subrealm level, these seven countries represented most of the African biomes, there are still numerous ecotypes and vegetation types that have not been adequately sampled [60–65]. On a local basis, we accounted for uncollected species with the Chao-I estimator [52], but if complete ecotypes were not included in the material studied, our estimation still tends to underestimate the actual diversity.

In addition, the probability of genera being taxonomically revised is not evenly distributed. Instead, there is a strong bias against the revision of species-rich genera [25], due to the complexity and resource-intensity of the task. Indeed, we found that larger genera also showed a stronger increase when comparing species numbers before and after recent revisionary work, and tried to correct for this bias by using a weighted multiplier. Furthermore, many revisions were conducted in the late 20th century based on morphology only. It is likely that integrative taxonomy will reveal further morphologically cryptic species in future revisions [66]. This suggests that the upper estimate is more adequate, since the underrepresented, species-rich genera are better accounted for. However, the fact that the really large genera of Darwin wasps all still await taxonomic revision means that we are likely still too low with our estimate.

Furthermore, to be able to apply the calculated multipliers to unrevised genera, it is necessary that at least some of their species in the Afrotropics are already described. However, we found that a large proportion of genera which are known to be widely distributed [26], did not have a single species recorded in the five focal countries, and we had to artificially assume a single species as a pre-revision representative (Table 2). Thus, for many megadiverse, enigmatic and often small-bodied subfamilies [25], the current state of taxonomy results in their underestimation in our approach. This limitation is especially visible in the Phygadeuontinae (Fig 5), which are completely off-radar in all focal countries except for Madagascar, where the subfamily has been treated before by Seyrig [67].

We encountered another possible bias during the extrapolation from focal countries to sub-realms using butterflies as faunistic surrogates. This approach relies on the completeness of the butterfly species records. However, due to limited accessibility, the species record of butterflies is, on average, more complete in the designated focal countries compared to other countries of the subrealm. Even though Papilionoidea together with Odonata [68, 69] are considered the best-studied insect groups in the Afrotropical region, only about 85–90% of the Afrotropical butterfly species are currently known (S. Safian, pers. comm.). Therefore, by extrapolating from focal countries to the subrealms using butterfly data, we tend to further underestimate the species richness of Darwin wasps.

In summary, every single step in our approach suffers from some underlying biases that likely led to an underestimation of the Afrotropical Darwin wasp diversity. Therefore, we conclude that the upper estimate obtained by using weighted multipliers (15,577 species) more closely reflects the true diversity of the megadiverse and severely understudied insect family Ichneumonidae in the Afrotropical realm, but that it likely still is an underestimate. It is noteworthy that this figure is close to the expert opinion of van Noort [58] (20,000 species), and encouraging that the use of expert opinion could add weight to extrapolations such as ours.

## No latitudinal distribution anomaly?

The current presented species richness estimation does not indicate a depletion in the species richness of Darwin wasps at lower latitudes. Instead, the Equatorial Afrotropics were estimated to be the most species-rich subrealm for Darwin wasps in the Afrotropical region (without correcting for area size), although closely followed by Madagascar & East African Coast (Table 3).

Overall, Darwin wasp diversity appears to be much higher in the tropics than previously thought [36, 37, 39]. Furthermore, we support that subfamily affiliation (Fig 5), biology (mode of parasitism) and ecological factors such as host distribution are more likely to explain species richness distribution in Darwin wasps than latitude itself [38, 70].

Although our estimate implies a massive under-description of the Afrotropical Darwin wasp diversity, we support Townes' observation [28] that the other tropical areas, namely the Neotropical and Indomalayan realms, are likely even richer. Indeed, a comparative taxonomic study between Peru and Uganda indicated that Rhyssinae are more species-rich in the former [71]. Whereas such direct comparisons that consider sampling effort are rare, species numbers from comparatively well-studied groups such as Ophioninae seem to point in the same direction [26, 54]. It would also match well with diversity patterns observed in other taxonomic groups, such as vascular plants, which are more than twice as species-rich in the Neotropics than in the similarly sized Afrotropics [31]. But further taxonomic investigations are required in tropical Darwin wasps to ascertain if this trend persists within other subfamilies and throughout the entire geographical realms.

## Handicaps and bottlenecks in taxonomic research

We here estimated that Darwin wasps in the Afrotropical realm represent a crucial proportion of the family's global diversity. It is imperative to understand this diversity on a systematic and taxonomic level, as an indispensable basis for applied studies in the fields of conservation and biocontrol. Due to inadequate funding, logistical constraints, and lack of inventory surveys in many Afrotropical countries, the collection and curation of (insect) samples, as well as species descriptions, experienced a drastic bottleneck for many decades [72, 73]. Many ecotypes and whole countries have never been adequately sampled for Darwin wasps and are severely underrepresented in natural history collections [61, 62, 74]. Furthermore, due to the colonialist history, important natural history collections are currently split between the former colonial power and the countries of origin, resulting in many logistic barriers both for local and international researchers [75]. To promote taxonomic research in Afrotropical countries and to accelerate the description rate, adequate funding and collaborative approaches are fundamental. While taxonomy is everywhere relatively poorly funded, at least there are researchers employed in many countries who work on taxonomy; these positions are particularly limited in Afrotropical countries. Future collaborations need to become more ambitious in both their geographic and taxonomic scope, and should focus on teaching and mentoring students in the systematics and taxonomy of Darwin wasps. A lack of young taxonomists specializing in this megadiverse family of insects, and a lack of jobs for them, is a major bottleneck in Darwin wasp research.

## Supporting information

**S1 File. Methodological pipeline.** R-script used to derive estimation from empirical data sets.
(R)

**S2 File. Morphospecies and revision factors.** List with extracted data and corresponding references.
(XLSX)

**S3 File. Afrotropical Darwin wasp checklist.** Extracted and updated from Yu et al. [26].
(CSV)

**S4 File. Afrotropical butterfly checklist.** Extracted and updated from Safian et al. [56].
(CSV)

**S5 File. Multiplier table.** Summary with all multipliers derived for the five Afrotropical sub-realms based on revisions, morphospecies or both.
(XLSX)

## Acknowledgments

This project was initiated as workshops during the Darwin wasp conferences held in 2019 in Basel and in 2022 at Station Linné in Öland. We would like to thank all participants of the workshops for their contribution to the initiation of the project. We also thank One Earth for the permission to reuse their illustration of the Afrotropical subrealms in this paper. We thank the editor of PLOS ONE and two anonymous reviewer for their constructive feedback on the manuscript.

## Author Contributions

**Conceptualization:** Noah Meier, Simon van Noort, Michal Rindos, Filippo Di Giovanni, Gavin R. Broad, Tamara Spasojevic, Andrew Bennett, Davide Dal Pos, Seraina Klopfstein.

**Formal analysis:** Noah Meier, Seraina Klopfstein.

**Funding acquisition:** Seraina Klopfstein.

**Investigation:** Noah Meier, Mikhaila Gordon, Simon van Noort, Terry Reynolds, Michal Rindos, Filippo Di Giovanni, Gavin R. Broad, Andrew Bennett, Seraina Klopfstein.

**Methodology:** Noah Meier, Mikhaila Gordon, Simon van Noort, Michal Rindos, Filippo Di Giovanni, Gavin R. Broad, Tamara Spasojevic, Andrew Bennett, Seraina Klopfstein.

**Supervision:** Tamara Spasojevic, Seraina Klopfstein.

**Visualization:** Noah Meier.

**Writing – original draft:** Noah Meier.

**Writing – review & editing:** Simon van Noort, Terry Reynolds, Michal Rindos, Filippo Di Giovanni, Gavin R. Broad, Tamara Spasojevic, Andrew Bennett, Davide Dal Pos, Seraina Klopfstein.

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
