## [Decision Letter · Decision Letter 0]

17 Jun 2024

PONE-D-24-19643Species richness estimation of the Afrotropical Darwin wasps (Hymenoptera, Ichneumonidae) based on complementary empirical dataPLOS ONE

Dear Dr. Meier,

Thank you for submitting your manuscript to PLOS ONE. After careful consideration, we feel that it has merit but does not fully meet PLOS ONE’s publication criteria as it currently stands. Therefore, we invite you to submit a revised version of the manuscript that addresses the points raised during the review process.

There was consensus among both reviewers that this is an improvement on previous estimates of Darwin wasp diversity in the region. While it would be desirable to acquire newer or "more complete" primary data for these estimates, the reviewers agree that this work brings something new and important to the field. I agree and predict that this manuscript will provide a new quantitative foundation for future estimates and assessments of diversity in this lineage and region. You will see that the reviewer comments are relatively minor in nature; please review them carefully as they largely address the clarity of the text and one figure.

We look forward to receiving your revised manuscript.

Kind regards,

Phillip Barden

Academic Editor

PLOS ONE

Journal Requirements:

"This project was initiated as workshop during the Darwin wasp conference in 2022 at Station Linné in Öland. We would like to thank all participants of the workshop for their contribution to the initiation of the project. We also thank One Earth for the permission to reuse their illustration of the Afrotropical subrealms in this paper. S. Klopfstein was supported by grant 310030_192544 of the Swiss National Science Foundation."

"SK was supported by grant 310030_192544 of the Swiss National Science Foundation. The funder played no role in study design, data collection and analysis, decision to publish, or preparation of the manuscript."

3. We note that [Figures 2 and 6] in your submission contain [map/satellite] images which may be copyrighted. All PLOS content is published under the Creative Commons Attribution License (CC BY 4.0), which means that the manuscript, images, and Supporting Information files will be freely available online, and any third party is permitted to access, download, copy, distribute, and use these materials in any way, even commercially, with proper attribution. For these reasons, we cannot publish previously copyrighted maps or satellite images created using proprietary data, such as Google software (Google Maps, Street View, and Earth). For more information, see our copyright guidelines: http://journals.plos.org/plosone/s/licenses-and-copyright.

a. You may seek permission from the original copyright holder of Figures 2 and 6 to publish the content specifically under the CC BY 4.0 license.  

Reviewers' comments:

Reviewer's Responses to Questions

**Comments to the Author**

1. Is the manuscript technically sound, and do the data support the conclusions?

Reviewer #1: Yes

Reviewer #2: Yes

2. Has the statistical analysis been performed appropriately and rigorously? 

Reviewer #1: Yes

Reviewer #2: N/A

3. Have the authors made all data underlying the findings in their manuscript fully available?

Reviewer #1: Yes

Reviewer #2: Yes

4. Is the manuscript presented in an intelligible fashion and written in standard English?

Reviewer #1: Yes

Reviewer #2: Yes

5. Review Comments to the Author

Reviewer #1: Meier et al used an empirical approach to estimate the species richness of ichneumonids in the Afrotropical region, and highlights the gaps between this new estimate with currently known species diversity. I think this is an innovative and important study for such a diverse group in an understudied region.

I do not have any major criticisms, although using butterflies, a distantly related groups of organisms with vastly different life history strategies/distribution pattern will undoubtedly skew the results here. I do not have a better alternatively suggestion as other parasitoids in the Afrotropical region is also vastly understudied and therefore cannot be used.

One other minor suggestion is to redo figure 2, as the dark red is rather hard to see on the map, and not colorblind-friendly with the red/green combination.

Reviewer #2: In this manuscript the authors estimate the species richness of Ichneumonid wasps in the Afrotropical region based on taxonomic literature, 2 museum collections, and extrapolations using better known butterflies. Using a somewhat complicated series of ratios and extrapolations starting with species lists from focal countries representing different geographic subrealms, they estimate that there are 9.2-15.6 K species of Ichneumonids in the Afrotropical region, a substantial increase over the 2.3 K currently documented species. The authors further argue that even their high estimate (employing weighted extrapolations of species based on genus size), is likely an underestimate due to the conservatism of their methods.

Overall, the paper is well written and interesting. The authors provide a nice background review of species estimates and diversity patterns for Braconids and Ichneumonids. They explain their methods reasonably well (e.g. ratios, multipliers, weights) and provide a flow chart, but I still found it somewhat difficult to follow everything they did and all of the assumptions made. It is a complicated process with a number of adjustments due to biases and lack of information (e.g., weighting, multiple levels/scales of extrapolation, exclusion of revisions with few specimens, using butterfly turnover as a proxy, etc.), and it’s a little difficult to understand what the influence of these various assumptions is on the final estimates. The authors justify their various decisions, methods and extrapolations well, and are generally conservative, but it is unclear how much confidence to have in their estimates. To a large extent this is a consequence of trying reasonably estimate diversity based on very limited and sparse data. Extrapolating richness estimates to the entire family based on numbers of undescribed morphospecies from a handful of genera of one subfamily seems particularly shaky. Despite the “shakiness” of this methodological architecture, built on a relatively weak foundation of data, it is clearly the best and most considered attempt to estimate the diversity of this important group of parasitoid wasps for the Afrotropics.

I do not have a lot of comments for improvement that don’t involve trying to accumulate more data (e.g. by sampling or more exhaustively surveying material in museums). Some of the assumptions or somewhat arbitrary decisions could be questioned, but as mentioned the authors justify their approaches and decisions pretty well. I found this to be an interesting manuscript that establishes an estimate of richness of this important clade that is far more reasoned and documented than the previous estimate (although that estimate is nicely nestled within the range of the current study ).

It might be worth mentioning that for the museum data and revisionary data where genetic methods were not used, true richness might be underestimated due to the presence of cryptic species.

Minor comments

L. 146 The statement “It remains to be determined whether the species lag in the Afrotropical region is attributable to study biases, such as a limited number of taxonomists working in this area..” seems a little like feigned ignorance – there seems to me little doubt that the number of taxonomists working in the Afrotropics is far far less than the Western Palearctic. I suggest the authors edit to “It remains to be determined how much of the species lag in the Afrotropical region…”

L. 160-161 more recent extensive sampling…

L. 162. Latitudinal gradient

L. 164 delete: Especially

L. 171 derived from

L. 253 Only revisions that approach the actual diversity of a genus…

L. 255 lowest specimen / species ratio. A list of the revisions…

L. 262 The ratio of the…

L. 264 The authors state “For series of morphospecies consisting only of undescribed species..” Are these morphospecies that appear to represent an undescribed genus – i.e. clusters of morphospecies that appear to be related but cannot be assigned to an existing genus? This should be stated.

L. 275 represented by only one..

L. 281 this sentence is a little confusing and could be restated for clarity

L. 309 “For revised genera, we kept the number of recorded species for each focal country.” What is meant by “kept” in this sentence?

L. 330 what is the “effective number”?

L. 332 A high turnover rate indicates little overlap in the species assemblage across the subrealms and a low turnover rate indicates extensive overlap.

L. 411 Ratios or generally phrased as “of” this and that, not between

L. 431 what again is the effective number? Is this just the sum of species known from each realm?

L. 473 delete even

L. 476 delete well

L. 478 include Seyrig reference

6. PLOS authors have the option to publish the peer review history of their article (what does this mean?). If published, this will include your full peer review and any attached files.

Reviewer #1: No

Reviewer #2: No

---

## [Editor Report · Decision Letter 1]

4 Jul 2024

Species richness estimation of the Afrotropical Darwin wasps (Hymenoptera, Ichneumonidae)

PONE-D-24-19643R1

Dear Dr. Meier,

We’re pleased to inform you that your manuscript has been judged scientifically suitable for publication and will be formally accepted for publication once it meets all outstanding technical requirements.

Kind regards,

Phillip Barden

Academic Editor

PLOS ONE
---

## [Editor Report · Acceptance letter]

19 Jul 2024

PONE-D-24-19643R1 

PLOS ONE

Dear Dr. Meier, 

I'm pleased to inform you that your manuscript has been deemed suitable for publication in PLOS ONE. Congratulations! Your manuscript is now being handed over to our production team.

Kind regards, 

on behalf of

Dr. Phillip Barden 

Academic Editor

PLOS ONE